# Perspectives on Active Transportation in a Mid-Sized Age-Friendly City: “You Stay Home”

**DOI:** 10.3390/ijerph16244916

**Published:** 2019-12-05

**Authors:** Irmina Klicnik, Shilpa Dogra

**Affiliations:** 1Faculty of Health Sciences (Community Health), University of Ontario Institute of Technology, Oshawa, ON L1G-0C5, Canada; Irmina.Klicnik@ontariotechu.net; 2Faculty of Health Sciences (Kinesiology), University of Ontario Institute of Technology, Oshawa, ON L1G-0C5, Canada

**Keywords:** physical activity, aging, social isolation, cycling, walking

## Abstract

Background: Active transportation is an affordable and accessible form of transportation that facilitates the mobility of older adults in their communities. Age-friendly cities encourage and support physical activity and social participation among older adults; however, they often do not adequately address active transportation. Our goal was to identify and understand the constraints to active transportation that older adults experience in order to inform the development of viable solutions. Methods: Focus group interviews were conducted with community dwelling older adults (n = 52) living in the City of Oshawa in Ontario, Canada; each focus group targeted a specific demographic to ensure a diverse range of perspectives were represented. Data were analyzed to identify themes; sub-group analyses were conducted to understand the experience of those from low socioeconomic status and culturally diverse groups. Results: Themes pertaining to environmental, individual, and task constraints, as well as their interactions, were identified. Of particular novelty, seemingly non-modifiable constraints (e.g., weather and personal health) interacted with modifiable constraints (e.g., urban design). Culturally diverse and lower socioeconomic groups had more favorable perspectives of their neighborhoods. Conclusion: While constraints to active transportation interact to exacerbate one another, there is an opportunity to minimize or remove constraints by implementing age-friendly policies and practices.

## 1. Introduction

Active transportation is a mode of transportation in which people get to places using their own power [1]. It is a form of physical activity wherein individuals travel or commute using modes such as walking or cycling. The purpose of the travel is not leisure, but work or task oriented. It is typically used for travel to school or work among children or young adults. Among older adults, active transportation can be used for travel associated with volunteer commitments, daily activities, or medical appointments. This form of transportation contributes significantly to daily physical activity levels and healthy aging [2]. Of utmost importance, it is a form of transportation that is both affordable and accessible.

Encouraging and enabling active transportation in cities may reduce the health burden and social isolation among older adults by facilitating mobility in the community, particularly among older adults who do not have access to a personal vehicle, or in cities where public transportation is suboptimal. However, the design of many communities does not support active transportation, particularly for older adults. The constraints to participating in active transportation are somewhat unique for older adults. For example, research indicates that, unlike young adults, older adults are not comfortable walking on snow-covered sidewalks [3], they require access to public toilets [4], and want benches or rest stops along their route [5]. Thus, the built environment and urban design are critical determinants of active transportation among older adults as it can either serve to limit or promote active transportation.

### 1.1. Transportation and the Age-Friendly Cities Guideline

To ensure that communities are designed with older adults in mind, the World Health Organization (WHO) put together a guideline on age-friendly cities in 2007 [6]. Many cities around the world have since committed to becoming age-friendly and have developed plans or strategies that have been approved by their municipal government. However, these age-friendly cities still lack significantly in the infrastructure necessary to support active aging and active transportation. Furthermore, active transportation is not directly addressed in the WHO guideline. Much of the emphasis in the *transportation* dimension is on public transportation, while the emphasis of the *outdoor spaces and buildings* dimension is on recreational physical activity. The latter addresses some important issues, however, there is a significant gap in supporting municipalities in creating age-friendly active transportation plans for their older residents.

To inform policies and practices pertaining to the design of age-friendly communities, we must first gather evidence of the issues experienced by older adults in their community. In a review of urban environments and age-friendly cities, Buffell and colleagues [7] cited several constraints that impact the lives of older adults. They suggested that there are many urban hazards that older adults must deal with in order to participate in life or to age in place. Importantly, they argued that older adults must be included in the development and maintenance of age-friendly environments. This may be particularly important in the transportation domain.

According to Shergold and colleagues, the mobility of older adults in their community is compromised by an increasing focus on the car, particularly in rural areas [8]. They also highlighted that many of the activities and services accessed by older adults are within short distances, making active transportation a feasible option. While previous commentaries have noted the need for better active transportation infrastructure [9,10], it is important to highlight again that active transportation is not part of the WHO age-friendly city guideline. As suggested by Lawler (2015), the scale required for transportation infrastructure to support adequate transportation options for older adults is grand, and will require significant investment. Alternatives to the car are needed to support active aging [10]. In fact, alternatives to public transit are also needed. In many cases, older adults cannot afford public transportation, and are thus unable to participate in activities of interest [11].

While some cities have included older adults in understanding transportation related concerns, the focus again has been on public transit [12]. Many cities have some form of public transportation, but some cities are too small to offer door-to-door services. In such cases, transportation becomes multi-modal because older adults may rely on active transportation to get to and from bus stops or train stations. Encouraging use of public transit leads to an increase in time spent in active transportation [13]. However, few plans are available to ensure safe and accessible active transport. We argue that in order for a city to be truly age-friendly, it must increase its focus on active transportation, not just public transit or recreational physical activity.

### 1.2. Theoretical Model

Work to date has investigated the many environmental constraints [5] as well as the individual constraints, such as fear of falling or health conditions such as arthritis [14], that older adults must negotiate when engaging in active transportation; however, little work has simultaneously assessed a broad array of constraints to understand the interaction between these constraints. This is critical for understanding how we can overcome constraints, and facilitate active transportation in older adults. This gap in knowledge can be addressed using a comprehensive theoretical approach as proposed by Newell [15]. The model is based on the concept that throughout the lifespan, certain constraints have more salience because of developmental stages. Older adulthood is characterized by developmental changes that affect functional independence, thus this model is useful in capturing a crucial stage of development in relation to an individual’s environment, and the demands of active transportation. Newell conceptualized optimal behavior and performance as a product of the interaction between three types of constraints: individual, environmental, and task. Individual constraints include structural factors (e.g., height, weight, and fitness) and functional factors (e.g., psychological qualities such as motivation and personality). Environmental constraints include geographical area, the physical environment, sociocultural environment, and policies. Finally, task constraints include the demands of an activity (e.g., ability to balance while performing a specific task) as well as equipment required (e.g., size of wheels on a gait aid such as a walker). The dynamic nature of Newell’s [15] model lends itself well to the study of constraints to active transportation because it recognizes that, although some constraints may be more salient for one individual than for another, the lack of one constraint type does not necessarily result in the desired outcome. Rather, it positions the outcome active transportation as a multifaceted problem, which requires multifaceted approaches for development of solutions. Of note, the literature also identifies promotors or facilitators of active aging and active transportation [16,17]. These may limit or counter constraints, but are not necessarily the inverse of constraints.

The purpose of this study was to understand the perspectives of a diverse group of older adults on the individual, environmental, and task constraints to participating in active transportation. While several studies have looked at specific types of constraints, that is, either environmental [4,5] or individual [14], our study is the first, to our knowledge, to investigate task constraints, as well as the interaction between constraints. This is an essential first step in identifying opportunities to properly address constraints. We were interested in speaking to a culturally diverse group of older adults from different socioeconomic environments to ensure that recommendations generated from this work would lead to the development of equitable solutions. Understanding these constraints is an important first step to helping cities support active transportation in older adults.

## 2. Materials and Methods

Study Design: This qualitative study used phenomenology, a method that is used to understand participant experiences [18]. A qualitative approach was necessary as we wanted to hear the voices of a diverse group of older adults whose voices are not often included in discussions pertaining to active transportation. Qualitative research is particularly important in gerontology as it helps clarify important issues related to the experience of aging [19]. Importantly, a survey of constraints would not provide an understanding of the constraints, nor does a validated or reliable survey of such nature exist. Nine focus groups were conducted with older adults, at separate locations, between October 2018 and January 2019. All methods and communications were approved by the Research Ethics Board at the University of Ontario Institute of Technology.

Participants: Community dwelling older adults (aged 55 years and older) were eligible to participate in this study. That is, older adults in assisted living facilities or long-term care were not included. Recruitment was done using several methods and through several contacts from the local senior’s community centers. Specifically, to ensure a representative sample of our city, we targeted “newcomer” social groups (n = 2, groups targeted at older adults who had recently immigrated to the country and/or did not speak English), low-income neighborhoods (n = 2, as per the Region of Durham Building on Health in Priority Neighbourhoods Report), and several different community locations including the library (n = 2), seniors centers (n = 2), and an apartment complex (n = 1). This resulted in a sample of 52 older adults. Participation in the study was voluntary, and all participants provided written informed consent.

Place: The focus groups were conducted with residents who lived in a mid-sized city (City of Oshawa) in southern Ontario, Canada. This city recently completed the consultation process to become an Age-Friendly designated city through the WHO. Approximately 1/3 of the population is over the age of 55 years, and there are five community centers in the city that specifically cater to this age group. The city is experiencing significant growth, and will need to prioritize active transportation to support urban mobility. Of note, there is a strong history of car manufacturing in the city which means much of the City’s identity and culture is based in cars. The culture is changing, and there is some political will to encourage active transportation.

Interview Guide: The purpose of these focus groups was to uncover and understand the constraints that older adults experience when engaging in active transportation. The focus group interview guide included an explanation of the research and purposes, a definition of active transportation, as well as questions that specifically probed participants about their current engagement in active transportation or their desire to engage in active transportation; their individual, environmental, and task constraints pertaining to active transportation; and their perception on whether their community supported active transportation. No personal, identifying data were collected from any of the participants; however, all participants met our eligibility criteria.

Research Team: The principal researcher and senior author of this paper has a PhD in kinesiology and health sciences with research expertise in the area of active aging. She and two research students conducted all of the focus groups. In some instances, a staff member was present because the participants were being recruited from a specific program. The first author on this paper was not present for data collection. She has expertise in the area of physical activity constraints, is a therapeutic recreation specialist with a bachelor’s degree in psychology, a master’s degree in health sciences, and several years of clinical experience working with older adults in hospital. The first and last author were responsible for all data analysis and interpretation.

Data Analysis: Audio tapes were transcribed verbatim and reviewed to ensure accuracy. Field notes were taken during the interview by a research student. Each focus group lasted roughly 30–45 min. Focus group transcripts were reviewed and analyzed by IK and SD.

A consensus approach was taken in the directed content analysis of the data. Directed content analysis was guided by processes described by Hsieh and Shannon [20]. A list of relevant concepts and terms was compiled based on existing literature on active transportation, and used as initial coding categories. Data analysis was completed manually by IK and SD. Codes were extracted independently in relation to each of the major areas outlined by the questions from the interview guide. From there, major themes were identified for each of the interview questions by collapsing codes. The research team discussed major themes and any disagreements were resolved during that time. As suggested by Miles and Huberman [21], major themes and corresponding codes were plotted into a flow chart, which identified relationships. The flow chart reflected themes identified during content analysis and identified underdeveloped areas that can be used to inform future research.

Themes were also compared between groups to better understand the differences between culturally diverse groups and groups of different socioeconomic backgrounds.

## 3. Results

Several themes arose from the data. These were: constraints (environmental, individual, and task) and the interactions between constraints, neighborhood perception, and promoters. Several subthemes arose during the content analysis of the nine focus group transcriptions. These themes and their respective subthemes are detailed below along with a comparison of the perspectives of those from different cultural and socioeconomic groups.

### 3.1. Theme 1: Constraints and Interactions

Environmental (Table 1), individual (Table 2), and task (Table 3) constraints that arose were consistent with previous literature. Sub-themes and sample quotes are provided in the respective tables.

Participants independently generated several unique interactions between constraints while describing their experience (Figure 1). These arose without explicit priming by focus group facilitators. First, the health or functional fitness related constraints (individual) such as arthritis were exacerbated by poor weather (environmental).

“In my case I’ve got a form of cerebral palsy, which I’ve had all my life and I fall just by looking at a patch of ice, let alone if there’s a slight rise in uh sidewalk, I trip if I’m not watching.”

Health also interacted with urban design factors such as sidewalks, crosswalk light timing, and lack of washroom facilities, or rest stops. Participants commented on health issues related to aging (e.g., arthritis, pain, and cognitive impairment) with their perception of difficulties navigating their neighborhood.

“So, 15 seconds or 19 seconds or 23 seconds to get from whatever that street is over to the mall is not enough for maybe somebody with a cane or a walker.”

Second, navigating bumpy or snow-covered sidewalks with gait aids was also mentioned, suggesting an interaction between environmental and task constraints. The environmental constraint here is two-fold: weather and sidewalk maintenance.

“I sometimes have to use a walker because I have arthritis my knees and in the-in the winter… lots of people don’t clear their snow properly its everyone’s responsibility, and the people at the corners especially. This is really a problem for a person with a walker.”

This sentiment was echoed by other participants, who also recalled stories of friends or neighbors who experienced similar constraints.

Finally, an interaction between individual and task constraints was mentioned when participants discussed their functional fitness in relationship to the height, seat size, and balance required on a bike. This was in the context of falls among older adults, where one participant noted that:

“At this stage [of life], you can’t afford to have a fall…a fall means a fracture, so we have to be very careful”.

The fear of falling was expressed by participants who were not using cycling as a mode of active transportation as well.

### 3.2. Promoters of Active Transportation

Some constraints raised were subsequently followed up with ways in which participants had negotiated the constraint; thus, participants made positive remarks about factors which facilitated their engagement in active transportation. Participants shared promoters related to the physical and built environment, health benefits, and a number of other factors.

Well maintained trails, the presence of local parks, and city-maintained gardens were all cited as municipality related promoters.

“ we are well provided for, but it’s just this niggling stuff”.

Participants also commented on using active transportation more frequently in the summer or in good weather. Further, some participants mentioned the health benefits of walking.

“If I go for a walk, I’m pain free for hours.”

Participants who used public transit spent more time in active transportation due to their walk to and from the bus stop as well as around the destination, and those who had a “buddy system” also engaged in more active transportation. Similarly, a lack of parking spaces forced some older adults to engage in active transportation.

### 3.3. Neighborhood Perception

Participants identified both positive and negative views of the neighborhood in which they lived. Some participants felt that their neighborhood in the city did not support active transportation and specifically said “we have nothing” or that it was fine for walking but not for cycling. They also mentioned that the infrastructure to support active transportation was not keeping up with urban sprawl and that despite these issues, they did not want to see an increase in their taxes. Those who felt positively about their neighborhood and its ability to support active transportation mentioned that most issues were relatively minor, and that overall the city was doing a good job. The differences in these views may be directly related to socioeconomic status and the neighborhood in which participants reside. For example, sidewalk quality was listed as a constraint in some groups, but others praised the infrastructure in their neighborhood.

### 3.4. Sub-Group Analysis

Participants in the low SES group generally espoused a more positive neighborhood perception with respect to infrastructure for active transportation, citing easy access to public transit and pride in their neighborhood. In some cases, their perspective was more negative. They cited perception of personal safety and a lack of residential snow clearing more frequently than other groups. Participants shared stories about incidents of crime that they had heard about from acquaintances, in the context of trails and walking paths being unsafe. One participant suggested:

“…the trails… [we] might be more comfortable knowing that if they were um patrolled or if it was a paved trail”.

Participants in the high SES group shared positive and negative neighborhood perceptions, acknowledging that, although they were generally “well provided for”, there were many improvements that needed to be made. Specifically, they cited policy issues as significant constraints to active transportation. These included policies related to infrastructure (wide, paved sidewalks), fines for drivers who fail to stop at a crosswalk or who park vehicles in cycling lanes, and the prospect of having to pay more taxes. As one participant, shared;

“…there seems to be a lot of inertia in city hall.”

Participants recruited from “newcomer” groups shared overall very positive neighborhood perceptions. They judged trails, accessibility (e.g., ramps and accessible entryways to buildings), community gardens and landscaping, and the presence of parks favorably. They frequently cited weather and traffic (secondary to road construction) related constraints, which was similar to other groups.

## 4. Discussion

Active transportation is an accessible and affordable mode of physical activity that facilitates active ageing. Supporting active transportation in older adults is critical for a city to be truly age-friendly. In fact, our data reveal that lack of support for active transportation prevented older adults from leaving their home, or participating in life. A novel finding of this study was that the constraints that older adults negotiate often interact with one another, exacerbating the primary constraint. It is clear from our data, that, to increase cycling as a mode of active transportation in older adults, several infrastructure investments, policy changes, and supports need to be in place. While walking is associated with fewer constraints, these constraints appear to affect a larger portion of the population. Our data reveal several opportunities for increasing active transportation in age-friendly cities.

Several constraints, such as weather and personal health, could be considered non-modifiable from the perspective of researchers and municipal governments. However, these constraints interacted with *modifiable constraints* such as urban design or skill, which enables development of interventions to increase active transportation. For example, older adults with physical impairments who required gait aids had additional constraints to walking in the winter due to sidewalks being covered in snow or ice. While it would appear that their own physical limitations are the primary constraint, this constraint can be removed or minimized by ensuring that sidewalks are properly maintained throughout the year. This can be dealt with in a number of ways by the municipality: they can make use of novel technologies and implement heated sidewalks, they can change policies and practices so that the onus of winter maintenance is not placed on residents, or they can create covered walking paths. Thus, seemingly non-modifiable constraints may be overcome through appropriate intervention.

This is an important consideration as cities work towards becoming age-friendly. It was clear from our data that urban design and the policies and practices of municipalities are key influencers of engagement in active transportation among older adults. This is in line with previous research that has also found sidewalk characteristics [22], lack of rest stops [23], and timing of lights at cross-walks or intersections [24,25] to be significant constraints to active transportation. A unique urban design constraint that was identified in this work was a lack of public phones. This is interesting in the context of safety. Many youth and adults have personal cellular phones, and thus feel safer when using trails or when out walking or cycling; however, many older adults have not fully adopted this technology, and rely upon pay phones. As cities continue to modernize, such utilities are being removed, and are having unintended consequences. In this case, removal of phones and the safety concerns arising may promote social isolation or reliance of cars [26].

Social isolation was an issue that arose through several focus groups. Participants indicated that fears associated with falling and traffic as well as lack of supportive infrastructure kept them at home. This is problematic as social isolation is a significant concern in our aging population. Estimates suggest that over 20% of older adults are socially isolated, with another 30% at risk [27]. Many age-friendly communities provide programming that encourages social participation of older adults, but are unable to provide appropriate access to active transportation, thus limiting participation. Active transportation, therefore, might be a critical counter measure to social isolation being experienced by older adults.

A novel and interesting aspect of the current work was the sub-analysis of those of lower SES and those who were from culturally diverse backgrounds. Interestingly, participants from newcomer groups were more satisfied with services and urban design compared to other groups. We also found that those in lower socioeconomic groups were more likely to have concerns around safety and urban design issues such as sidewalks, but had more positive overall perceptions of their neighborhood. On the other hand, older adults of higher socioeconomic backgrounds have more concerns with municipal policies and practices. This may be due to a higher level of volunteerism or their own career backgrounds, increasing their awareness and understanding of the importance of municipal workings [28]. The social capital garnered through regular or incidental interactions with neighbors or one’s community while engaging in active transportation has also been shown to increase health-promoting behavior [29]. People who live in neighborhoods which promote active transportation have a greater sense of social capital than those who live in car-centric communities [30,31]. Participants in the low socioeconomic groups were the only groups who commented that bus use contributed to their active transportation, which may explain their more positive neighborhood perceptions in the context of social capital gained through this mode.

There are some strengths and weaknesses of this work that should be considered. First, we used a diverse group of older adults from a variety of neighborhoods and formal groups that allowed for important sub-group analyses, and provide a representative sample of our city. Our sample size was also sufficient to reach saturation, and all data were collected over the fall and winter seasons. Some weaknesses of the current study are that we did not include older adults living in assisted living or retirement residences. Isolated older adults living in the community who do not access community centers were also not included. In addition, actual time spent engaging in active transportation was not considered due to the scope of this study, nor was perceived level of loneliness or isolation. Finally, we were unable to include Indigenous Canadians in our study due to the timelines and approval requirements. Future research should consider the experience of older Indigenous Canadians to better understand their experience of active transportation and age-friendly cities.

### Implications and Recommendations

This work provides novel insights into active transportation and age-friendly communities. First, based on our findings that constraints to active transportation facilitated social isolation, it is clear that communities aiming to be truly age-friendly must address active transportation in their plans. Second, the interactions identified clearly indicate that many of the non-modifiable constraints to active transportation, such as weather and functional capacity, are exacerbated by policies and practices of local municipalities. Thus, an age-friendly and accessibility lens needs to be applied to all working committees in municipalities. Finally, the differences observed between neighborhoods indicates that there is an inequity in the way municipalities prioritize infrastructure related to active transportation. To ensure accessibility across the diverse range of older adults, more equitable policies and practices need to be put in place.

Based on our work, we would recommend that the WHO Age-Friendly guideline be updated to include active transportation. This would ensure that cities and communities around the world who are interested in the age-friendly designation be required to undertake significant consultation in their communities to better understand the needs of their residents. Given that many cities around the globe are declaring climate emergencies, and there is increasing concern about the health effects of climate change and air pollution [32], an increased emphasis on active transportation is both timely and necessary. We would also recommend that municipalities include active transportation in their master transportation plans, and consult with older adults when working on these plans. According to statistics from major cities, older adults are disproportionally the victims of traffic related fatalities [33], and many cities have unknowingly promoted victim blaming strategies to help older adults in reducing these fatalities. For example, police and municipalities have encouraged pedestrians to wear reflective clothing, remove earphones, and make eye contact with drivers while walking or cycling, instead of putting in appropriate infrastructure that ensures the safety of those engaging in active transportation. This is an indicator of the car-centric culture of cities. Finally, it is clear that urban design is critical for facilitating active transportation. Thus, municipal staff in charge of urban planning need to develop more age-friendly policies. For example, new developments could be required to ensure appropriate multi-use pathways, protected bike lanes, and rest stops and benches. While it requires significant finances to retrofit old neighborhoods, no new car-centric neighborhoods should be created moving forward.

Ultimately, we found that active transportation is central to an age-friendly city, and needs to be more carefully considered by the WHO as well as local municipalities. Future research comparing health and quality of life of older adults living in communities that support active transportation to those who do not is needed to determine the magnitude of the impact of active transportation.

## 5. Conclusions

Focus group data from a diverse group of older adults indicate that they negotiate several individual, environmental, and task related constraints to participating in active transportation, and, while these constraints interact to exacerbate one another, there is an opportunity to minimize or remove constraints by implementing age-friendly policies and practices. For a city to be truly age-friendly, active transportation must be prioritized.

## Figures and Tables

**Figure 1 ijerph-16-04916-f001:**
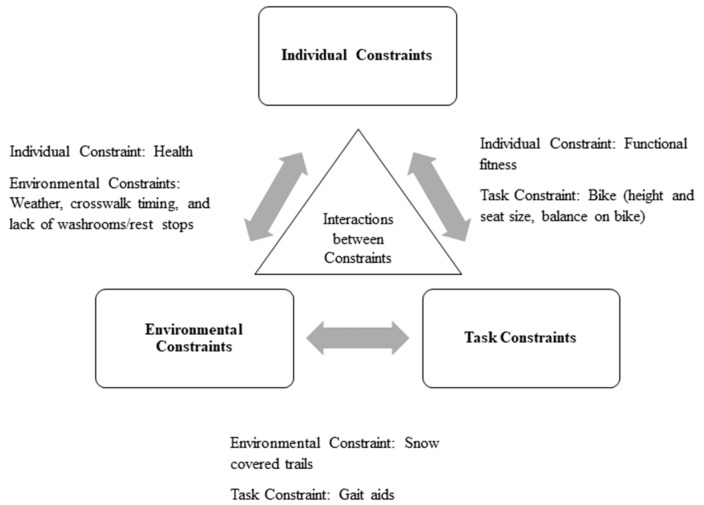
A depiction of the interaction between the constraints to active transportation in older adults.

**Table 1 ijerph-16-04916-t001:** Environmental Constraints: Subthemes, examples and supporting quotations.

Subthemes	Examples Cited	Sample Quotations
Weather	Ice, humidity, cold, rain, snow, and slippery conditions	Like, uh, some places there’s too slippy I think and then one has to sort of be careful.
Sidewalks and Roads	Lack of residential snow clearing Quality and width of sidewalk/roads/bike lanes Lack of curb cuts Increased traffic and construction Lack of bike lanes Lack of cross-walks Insufficient length of pedestrian traffic lights	When you get to the winter time the city relies on every individual...to clear their path... But if you ever try walking down them [sidewalks], you’ll find that not everyone does, right? Walking is too dangerous. They’ll do a bike lane by painting a line, but that doesn’t keep the bicycles and cars separate
Urban Design	Car-centric Lack of bike racks Continuity of bike trails Lack of rest stops/benches Lack of phone booths Lack of washrooms/water fountains Distance to stores/services	Um I was wondering how practical it was to um put out some rest stops, you know perhaps a bench or something like that for people who can walk short distances, but just need to stop and-and you know? I think a lot of it is some drivers, car drivers don’t accept bikes on the road. They just...I mean you see them go around and they just cut you off. So, I get to the grocery store, where do I put my bike?
Transit	Accommodation for mobility issues Cleanliness of shelters and stops Lack or benches/shelters Connectivity of routes	What we need and not necessarily a full size bus that uses a lot of gas, but a small bus say coming every hour right outside ABC Street, not upon XYZ Road or anywhere else. We need it outside the door. And like maybe a small bus that would take us to doctors appointments, clinics, hospital for appointments and stuff.
City/regional policies and practices	Cars parked in bike lanes Lack of enforcement (snow removal, construction, driving through cross-walks) Trail maintenance (e.g., garbage, animal control)	And they [buses]- they’ll sit there for 20–25 min, and that’s right in the bike lane, and those are region buses by the way. And if you have a car coming, done, you’re done, you’re toast…I just talked to an officer. He says-he says unless it specifically states on those streets where you have the bike lanes for bicycle, that is enforceable you know, no parking on the bicycle lanes, they don’t enforce it. So this at your own discretion, take your chances you know.

**Table 2 ijerph-16-04916-t002:** Individual Constraints: Subthemes, examples and supporting quotations.

Subthemes	Examples Cited	Sample Quotations
Health	*Previous surgeries* *Cognitive impairment* *Pain* *Arthritis* *Herniated disc*	Arthritis slows you down. As you get older you know, your abilities go down…
Fear of falls/injury	Functional fitness Risk of injury Embarrassment Social Isolation	At this stage you can’t afford to have a fall, a fall means a fracture or something…so we have to be very careful. I won’t even ride my bike on the road anymore. To me, its just too scary. You stay home.
Perception of Personal Safety	Presence of drugs/illicit activity on trails Lack of security patrols Wild animals Lack of lighting and phone booths Poor signage	Drunks, men [people] that are yelling at the top of their voices they’re swearing they’re…scary. The trails, um, they might be more comfortable knowing that they were um patrolled if it was a paved trail.
Functional Fitness	Balance Agility Mobility Fatigue Strength	I’m afraid of falling that’s…a big one for me. Balance, my balance is bad and a lot of us have the same problem. I can’t get my leg over the seat to get on the bicycle...
Financial	Cost of bicycle	What about financial barrier to um cycling? What if I don’t have a bike or I don’t have...and I have to get that stuff.
Personal Characteristics	Lazy Body Lack of knowledge of bike use/safety	Well I’m active at home, but I am too lazy to walk, so in the summer during the days when its hard to get parking over here... I would walk.

**Table 3 ijerph-16-04916-t003:** Task Constraints: Subthemes, examples and supporting quotations.

Subthemes	Examples Cited	Sample Quotations
*Task* *Walking related*	Gait aids not optimized for outdoor/all terrain use Difficulty with some tasks	I um sometimes have to use a walker because I have arthritis my knees and in the-in the winter, I live in a residential neighbourhood in ABC and in the winter lots of people don’t clear their snow properly its everyone’s responsibility, and the people at the corers especially. This is really a problem for a person with a walker. ...going to the grocery store, and you have a couple of bags, well it’s, I’d rather take my car because of the strength issue involved there.
*Cycling related*	Height of bike seat Size of seat Balance required to ride	There’s all kinds of places I can learn how to drive a car, where do I learn how to drive a bike.

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
