# Peer review of "Perspectives on Active Transportation in a Mid-Sized Age-Friendly City: “You Stay Home”"

_ijerph, 2019, doi:10.3390/ijerph16244916_

Round 1

Reviewer 1 Report

Dear authors,

your study used a qualitative approach to identify and understand constraints to active transportation in older adults. Nine focus group interviews were conducted with 51 older adults. The study gave interesting new insights, because it investigated the interaction between different constraints (environmental, individual, task). Nevertheless, there are some issues, which should be improved to strengthen the paper.

Introduction:

In my opinion the introduction did not clarify, why your research question required a qualitative research design (focus groups). "Interactions between constraints” could also be assessed with a quantitative design. What is the reason and advantage of your design? Explain the rationale of your study design.

some minor issues and a suggestion.

Line 72: (1986) should be deleted

Line 79: Please give references for the statement that “several studies have looked at specific types of constraints …” and you may explain (in short) on which type of constraints these studies have been looked at.

As a suggestion because I’m not familiar with the model of Newell: Why did you choose this model? Due to your reference-list this model is about motor development in children. Why didn't you refer to models, which especially refer to the interaction of environments with older adults, maybe the "Competence-(environmental) Press Model" of Lawton and Nahemow's (1973) or other theoretical assumptions in the field of Environmental Psychology. Most of these models (like the competence-press model) assume that behavioral outcomes (e.g. active transportation) vary as a function of personal competence and environmental press. Environmental press are environmental demands (context factors) that have an impact on the behavior of individuals. This impact is mediated by the competence (or performance potential) of the individuum. In my opinion this would fit very well as theoretical background for your study and such a theoretical assumption gave an idea about how the different constraints may interact.

Materials and Methods

Line 97: Please proved descriptive details of the focus-group members (age: range, mean, SD; gender) and please specify how many focus groups have you conducted for which kind of groups (e.g. low-income neighborhoods, etc.).

Line 107: I would place these information about the contributions of the authors at the end of the paper and not within the result section

Results

The interesting point of your paper is the analysis of the interactions between different constraints within different groups. It would be helpful for the reader, if you could provide an overview (maybe a graphic) how the constraints and which of them interact.

Line 133: Define “promoters” in the theoretical introduction and not at the end of the result section. Promoters seem to be a construct for your analysis and should be introduced theoretically.

Line 199: Please go into detail here, because these results are interesting: How were these differences be related to socioeconomic status. And concerning the following sentence: Did you mean that within the same neighborhood some groups (which one) listed sidewalks as a constraints and others (which group) not? Please specify, if it is the same neighborhood and which groups specified which constraints.

Line 204: I think it should be “positive and negative perceptions” because this group mentioned a “lack of residential snow clearing and reported (shared) stories about incidents of crime”.

Table 2: I didn’t get why you put “social isolation” to the subtheme of “fears of falls/injury”? Besides fears of falls, there are certainly more reasons for older adults to feel isolated / lonely.

Discussion

Line 228: What do you mean with "another" novel finding? The first mentioned finding (lack of support prevented older adults from leaving their home) is not "novel".

Line 253: Concerning the important issue of “safety-perception”, you could add the realist synthesis of Yen et al., 2014.
Yen et al. (2014) How design of places promotes or inhibits mobility of older adults: realist synthesis of 20 years of research. Journal of Aging and Health. DOI: 10.1177/0898264314527610

Author Response

Your study used a qualitative approach to identify and understand constraints to active transportation in older adults. Nine focus group interviews were conducted with 51 older adults. The study gave interesting new insights, because it investigated the interaction between different constraints (environmental, individual, task). Nevertheless, there are some issues, which should be improved to strengthen the paper.

Introduction: In my opinion the introduction did not clarify, why your research question required a qualitative research design (focus groups). "Interactions between constraints” could also be assessed with a quantitative design. What is the reason and advantage of your design? Explain the rationale of your study design.

RESPONSE: Thank you for bringing this up. We have added a justification to the methods section of the paper that reads as follows:

“A qualitative approach was necessary as we wanted to hear the voices of a diverse group of older adults whose voices are not often included in discussions pertaining to active transportation. Qualitative research is particularly important in gerontology as it helps clarify important issues related to the experience of aging[15]. Importantly, a survey of constraints would not provide an understanding of the constraints, nor does a validated or reliable survey of such nature exist.”

some minor issues and a suggestion. Line 72: (1986) should be deleted

RESPONSE: Thank you, this has been deleted.

Line 79: Please give references for the statement that “several studies have looked at specific types of constraints …” and you may explain (in short) on which type of constraints these studies have been looked at.

RESPONSE: We have added references as well as information on which category of constraints we are referring to. Each of these constraints were described in the preceding paragraphs.

As a suggestion because I’m not familiar with the model of Newell: Why did you choose this model? Due to your reference-list this model is about motor development in children. Why didn't you refer to models, which especially refer to the interaction of environments with older adults, maybe the "Competence-(environmental) Press Model" of Lawton and Nahemow's (1973) or other theoretical assumptions in the field of Environmental Psychology. Most of these models (like the competence-press model) assume that behavioral outcomes (e.g. active transportation) vary as a function of personal competence and environmental press. Environmental press are environmental demands (context factors) that have an impact on the behavior of individuals. This impact is mediated by the competence (or performance potential) of the individuum. In my opinion this would fit very well as theoretical background for your study and such a theoretical assumption gave an idea about how the different constraints may interact.

RESPONSE: The reviewer brings up an interesting point, and an interesting model. Newell’s model was chosen due to the inclusion of task constraints; something we have not seen in other models looking at the interaction between the environment and health/health behaviour. Task constraints are important to understand in an older population.

The reviewer is correct that this model was originally conceptualized to explain motor development and coordination, however, the theory has since proven highly useful and adaptable to other areas. Specifically, for organizing systematic reviews, and for understanding athlete development, and behavioural outcomes (some references provided below). We are not aware of this model being used in an aging population or in the context of active transportation; however, we were unable to find any similar models that have been used in these contexts. As such, Newell’s model informed our hypotheses and the development of our interview guide.

Rienhoff, R., Tirp, J., Strauß, B., Baker, J., & Schorer, J. (2016). The ‘quiet eye’ and motor performance: a systematic review based on Newell’s constraints-led model. Sports Medicine, 46, 589-603. Webdale, K., Baker, J., Schorer, J., & Wattie, N. (2019). Solving sport’s ‘relative age’ problem: A systematic review of proposed solutions. International Review of Sport and Exercise Psychology. Phillips, E., Davids, K., Renshaw, I., & Portus, M. (2010). Expert performance in sports and the dynamics of talent development. Sports Medicine, 40, 1–13. Materials and Methods: Line 97: Please proved descriptive details of the focus-group members (age: range, mean, SD; gender) and please specify how many focus groups have you conducted for which kind of groups (e.g. low-income neighborhoods, etc.).

RESPONSE: We did not gather personal information from focus group attendees, and allowed them to participate in the sessions as long as they met our eligibility criteria. We have added further detail regarding this as well as the different groups from whom data were collected. The edited and newly included lines read as follows:

“Specifically, to ensure diversity, we targeted ‘newcomer’ social groups (n=2, groups targeting older adults that had recently immigrated to the country and/or didn’t speak English), low-income neighborhoods (n=2), and several different community locations including the library (n=2), seniors centers (n=2), and an apartment complex (n=1).”

“No personal, identifying data were collected from any of the participants, however all participants met our eligibility criteria.”

Line 107: I would place these information about the contributions of the authors at the end of the paper and not within the result section.

RESPONSE: We included this information in accordance with CORE-Q guidelines but are happy to remove it if the editors feel it should be removed.

Results: The interesting point of your paper is the analysis of the interactions between different constraints within different groups. It would be helpful for the reader, if you could provide an overview (maybe a graphic) how the constraints and which of them interact.

RESPONSE: Thank you, this is a great suggestion! We have added a figure depicting the interactions observed.

Line 133: Define “promoters” in the theoretical introduction and not at the end of the result section. Promoters seem to be a construct for your analysis and should be introduced theoretically.

RESPONSE: Thank you for the suggestion. We have added this information to the introduction of the paper.

Line 199: Please go into detail here, because these results are interesting: How were these differences be related to socioeconomic status. And concerning the following sentence: Did you mean that within the same neighborhood some groups (which one) listed sidewalks as a constraints and others (which group) not? Please specify, if it is the same neighborhood and which groups specified which constraints.

RESPONSE: We targeted participants from neighbourhoods that are identified as “priority” based on SES and health indicators (Building on Health in Priority Neighbourhoods, Region of Durham, 2015). We did not gather individual sociodemographic data and thus are using each group as a representative of low vs. higher SES.

We have tried to clarify this in the methods section of the paper. The edited lines read as follows:

“Specifically, to ensure a representative sample of our city, we targeted ‘newcomer’ social groups (n=2, groups targeted at older adults that had recently immigrated to the country and/or didn’t speak English), low-income neighborhoods (n=2, as per the Region of Durham Building on Health in Priority Neighbourhoods Report), and several different community locations including the library (n=2), seniors centers (n=2), and an apartment complex (n=1).”

Line 204: I think it should be “positive and negative perceptions” because this group mentioned a “lack of residential snow clearing and reported (shared) stories about incidents of crime”.

RESPONSE: We agree and have edited the subsequent sentence.

Table 2: I didn’t get why you put “social isolation” to the subtheme of “fears of falls/injury”? Besides fears of falls, there are certainly more reasons for older adults to feel isolated / lonely.

RESPONSE: The issue of social isolation came up during discussions about falls, therefore, we included it in this category. While we agree with the reviewer that social isolation can be a result of many factors, in our focus groups and in the context of active transportation, it seems to be driven by fear of falling.  

Discussion: Line 228: What do you mean with "another" novel finding? The first mentioned finding (lack of support prevented older adults from leaving their home) is not "novel".

RESPONSE: Thank you, we have edited this line.

Line 253: Concerning the important issue of “safety-perception”, you could add the realist synthesis of Yen et al., 2014.
Yen et al. (2014) How design of places promotes or inhibits mobility of older adults: realist synthesis of 20 years of research. Journal of Aging and Health. DOI: 10.1177/0898264314527610

RESPONSE: We have added this to the discussion.

Reviewer 2 Report

Kia ora

Thanks for the opportunity to review your well written paper; it is an interesting contribution to the literature.

I suggest you modify your title and abstract to include where the study took place this key information is missing until  indirectly mentioned in the method.

Ln 42 - surely personal safety and crime would also be factors?

Ln 92 what do you mean by community dwelling?

Ln 95 did you recruit Indigenous participants to this study? if not why not?

Ln 107 Great transparency and clear contributions to the paper.

The tables are a clear and easy way to present some of the data. 

Given you claim to collected a diverse sample of participants it would have been good to see  an ethnic analysis.

This paper would have benefitted by being more clearly located in a particular geographic area. The demographics socio-political context of where data collected has not been disclosed.

Address some of these matters I think you will have a stronger paper.                                                             

Author Response

Kia ora. Thanks for the opportunity to review your well written paper; it is an interesting contribution to the literature. I suggest you modify your title and abstract to include where the study took place this key information is missing until  indirectly mentioned in the method.

RESPONSE: Thank you for this suggestion, we have added this information to the title and abstract.

Ln 42 - surely personal safety and crime would also be factors?

RESPONSE: Yes, indeed they appear to be based on our findings; however, from the literature it appears that these are constraints among youth and adults as well. In this line we were trying to highlight constraints unique to older adults.

Ln 92 what do you mean by community dwelling?

RESPONSE: By community dwelling we mean that the participants were living in the community, not in assisted living or long-term care. We have clarified this in the methods section.

Ln 95 did you recruit Indigenous participants to this study? if not why not?

RESPONSE: We had hoped to include indigenous older adults in our study; however, in Canada, the process for ethics approval for research with indigenous Canadians requires documentation, letters of support, and approval of indigenous leaders. We were unable to obtain these approvals in the timelines for our research grant. This was my third attempt at including indigenous Canadians in research, and I was once more unable to do so. It is disappointing, and definitely a limitation. We have added the following line to the manuscript’s discussion section.

“Finally, we were unable to include Indigenous Canadians in our study due to the timelines and approval requirements. Future research should consider the experience of older Indigenous Canadians to better understand their experience of active transportation and age-friendly cities.”

Ln 107 Great transparency and clear contributions to the paper. The tables are a clear and easy way to present some of the data. 

RESPONSE: Thank you!

Given you claim to collected a diverse sample of participants it would have been good to see  an ethnic analysis.

RESPONSE: We have made some changes to our methods section to clarify our recruitment process. We made every effort to end up with a representative sample, by targeting recruitment from specific groups and neighbourhoods. However, we did not ask individual participants for any personal or identifying information. According to our records which simply asked participants (yes/no) if they were 55 and older, and whether they were an immigrant to Canada at any time, or a visible minority, 14% of the sample was a visible minority and 14% reported that they immigrated to Canada; however, many participants did not respond to these questions.

Nevertheless, in section 3.4 we include a comparison of the newcomer groups to the rest of the sample which represents differences between a culturally diverse group and a Caucasian-non-immigrant group.

This paper would have benefitted by being more clearly located in a particular geographic area. The demographics socio-political context of where data collected has not been disclosed. Address some of these matters I think you will have a stronger paper.

RESPONSE: We have added some details about the geographical location and socio-political context to the methods. The new lines read as follows:

“Place: The focus groups were conducted with residents who lived in a mid-sized city (City of Oshawa) in southern Ontario, Canada. This city recently completed the consultation process to become an Age-Friendly designated city through the WHO. Approximately 1/3rd of the population is over the age of 55 years, and there are five community centers in the city that specifically cater to this age group. The city is experiencing significant growth, and will need to prioritize active transportation to support urban mobility. Of note, there is a strong history of car manufacturing in the city which means much of the City’s identity and culture is based in cars. The culture is changing, and there is some political will to encourage active transportation.”

Round 2

Reviewer 1 Report

Dear author(s),

you did a great job in improving the first version. Your second submission is clearly written and the issue should be relevant and interesting for readers of IJERPA.

Author Response

Thank you.